# *Cutibacterium* spp. Infections after Instrumented Spine Surgery Have a Good Prognosis Regardless of Rifampin Use: A Cross-Sectional Study

**DOI:** 10.3390/antibiotics12030518

**Published:** 2023-03-04

**Authors:** Susana Núñez-Pereira, Eva Benavent, Marta Ulldemolins, Beatriz Sobrino-Díaz, José A. Iribarren, Rosa Escudero-Sánchez, María Dolores Del Toro, Andrés Nodar, Luisa Sorli, Alberto Bahamonde, Helem H. Vilchez, Oriol Gasch, Elena Muñez, David Rodríguez-Montserrat, María José García-País, Sleiman Haddad, Julia Sellarès-Nadal, Oscar Murillo, Dolors Rodríguez-Pardo

**Affiliations:** 1Spine Unit, Orthopaedic Surgery Department, Vall d’Hebron University Hospital, 08035 Barcelona, Spain; 2Infectious Diseases Department, Hospital Universitari de Bellvitge, IDIBELL, Universitat de Barcelona, L’Hospitalet de Llobregat, 08907 Barcelona, Spain; 3Department of Infectious Diseases, Hospital Regional Universitario Málaga, 29011 Málaga, Spain; 4Infectious Diseases Department, Hospital Universitario Donostia, 20014 Gipuzkoa, Spain; 5Infectious Disease Department, University Hospital Ramón y Cajal, 28034 Madrid, Spain; 6CIBERINFEC, ISCIII-CIBER de Enfermedades Infecciosas, Instituto de Salud Carlos III, Av. de Monforte de Lemos, 5, 28029 Madrid, Spain; 7Infectious Diseases Unit, Hospital Universitario Virgen Macarena, 41009 Seville, Spain; 8Departamento de Medicina, Instituto de Biomedicina de Sevilla (IBiS), Universidad de Sevilla, 41009 Seville, Spain; 9Infectious Diseases Unit, Internal Medicine Department, 36312 Vigo, Spain; 10Instituto de Investigación Biomédica Galicia Sur, Hospital Álvaro Cunqueiro, 36312 Vigo, Spain; 11Infectious Diseases Department, Hospital del Mar, 08003 Barcelona, Spain; 12Infectious Pathology and Antimicrobials Research Group (IPAR), Institut Hospital del Mar d’Investigacions Mèdiques (IMIM), CEXS-Universitat Pompeu Fabra, 08003 Barcelona, Spain; 13Department of Internal Medicine-Infectious Diseases, Hospital Universitario del Bierzo, 24411 Ponferrada, Spain; 14Infectious Diseases Unit, Department of Internal Medicine, Hospital Universitario Son Espases, 07120 Palma de Mallorca, Spain; 15Infectious Diseases Department, Hospital Parc Tauli de Sabadell, University Autonoma of Barcelona, 08208 Barcelona, Spain; 16Infectious Diseases Unit, Internal Medicine Department, Hospital Puerta de Hierro-Majadahonda, 28222 Madrid, Spain; 17Orthopedic Surgery Department, Germans Trias i Pujol University Hospital, 08916 Badalona, Spain; 18Infectious Disease Unit and Microbiology Departments, Hospital Universitario Lucus Augusti, 27003 Lugo, Spain; 19Infectious Diseases Department, Vall d’Hebron, Hospital Universitari Vall d’Hebron, 08035 Barcelona, Spain; 20Medicine Department, Universitat Autònoma de Barcelona, 08193 Bellaterra, Spain

**Keywords:** surgical site infection, spine surgery, *Cutibacterium* spp., rifampin

## Abstract

Infection after spinal instrumentation (IASI) by *Cutibacterium* spp. is being more frequently reported. The aim of this study was to analyse the incidence, risk factors, clinical characteristics, and outcome of a *Cutibacterium* spp. IASI (CG) compared with non-*Cutibacterium* IASI (NCG) infections, with an additional focus on the role of rifampin in the treatment. All patients from a multicentre, retrospective, observational study with a confirmed IASI between January 2010 and December 2016 were divided into two groups: (CG and NCG) IASI. Baseline, medical, surgical, infection treatment, and follow-up data were compared for both groups. In total, 411 patients were included: 27 CG and 384 NCG. The CG patients were significantly younger. They had a longer median time to diagnosis (23 vs. 13 days) (*p* = 0.025), although 55.6% debuted within the first month after surgery. *Cutibacterium* patients were more likely to have the implant removed (29.6% vs. 12.8%; *p* = 0.014) and received shorter antibiotic regimens (*p* = 0.014). In 33% of *Cutibacterium* cases, rifampin was added to the baseline therapy. None of the 27 infections resulted in treatment failure during follow-up regardless of rifampin use. *Cutibacterium* spp. is associated with a younger age and may cause both early and late IASIs. In our experience, the use of rifampin to improve the outcome in the treatment of a *Cutibacterium* spp. IASI is not relevant since, in our series, none of the cases had therapeutic failure regardless of the use of rifampin.

## 1. Introduction

In recent years, there has been an increasing awareness of the relevance of *Cutibacterium* spp. as a causal agent of implant-associated surgical site infection (SSI) [1]. Infections caused by this microorganism are characteristically of low virulence, and the absence of fever, drainage, or changes in laboratory tests make their diagnosis difficult [2]. However, the diagnosis of these infections has become increasingly frequent. This can probably be attributed, at least in part, to an improved diagnosis procedure involving a prolonged incubation time of the samples obtained for culture and the sonication of the removed implants when available [3].

Regarding the aetiology, early infections after spine instrumentations (IASIs) are often caused by virulent and aggressive microorganisms such as *Staphylococcus aureus* and Enterobacterales, and it is not uncommon for them to be polymicrobial [4]. On the contrary, late IASIs are typically caused by less virulent bacteria such as *Staphylococcus epidermidis* or *Cutibacterium* spp., often as monomicrobial infections [5,6].

*Cutibacterium* spp. is a facultative anaerobic Gram-positive bacillus, a commensal organism of the skin with a predilection for pilosebaceous follicles and sebaceous glands. In most cases, it is considered non-pathogenic or contaminating. Due to its preference for areas with high concentrations of sebaceous glands, it is present mostly in the axilla, groin, and upper back [7,8]. However, it has been identified as a causative agent of implant-related infections, especially in the shoulder region. C acnes could be responsible for 25–40% of prosthetic shoulder infections [7,8,9]. It is rather uncommon in hip and knee prosthesis infections [9,10], but its presence in the upper back region makes it likely a causative agent of IASI, as previous studies have suggested [11].

In our experience [12], *Cutibacterium* spp. constitutes 4.7% of the causative isolates among 411 IASIs reviewed, and its frequency ranges from 3.5% in infections diagnosed during the first month after surgery to 15.6% in those diagnosed after the third month. Previous studies regarding risk factors for IASIs have shown that they occur more frequently in elderly patients with lumbar or lumbosacral instrumentations [13,14,15], although these characteristics are not met in *Cutibacterium* spp. infections. Grossi et al. [11] analysed the risk factors associated with *Cutibacterium* spp. IASIs and concluded that a younger age, lower BMI, and thoracic instrumentation were the most relevant variables. To the best of our knowledge, this is the only paper that exclusively addresses IASIs due to *Cutibacterium* spp.; thus, further studies are needed to address not only the risk factors associated with *Cutibacterium* spp. IASIs, but also different aspects in relation to medical treatment, such as the impact of the use of rifampin in its outcome. Rifampin is not stand-alone therapy but is prescribed in combination with other antibiotics. Some authors justify the use of rifampin in the treatment of *Cutibacterium* spp. IASIs by the theorical superiority of rifampin-based regimens in terms of greater cure rates and a lower percentage of infection recurrences due to its antibiofilm activity, which is particularly relevant if the implant has not been removed [16]. However, other authors have failed to demonstrate the benefit of rifampin use in *Cutibacterium* spp. IASIs [17]. Therefore, the role of rifampin in these infections remains uncertain. Addressing the modifiable risk factors associated with *Cutibacterium* spp. IASIs, as well as optimizing their medical and surgical management, would improve their prognosis and enable the development of better preventive strategies specifically aiming to protect the population at risk.

We hypothesised that the clinical presentation of an IASI caused by *Cutibacterium* spp. differs from the clinical presentation of patients with an IASI caused by other microorganisms, and that rifampin use can improve their prognosis, particularly in cases treated with implant retention. The aim of this study was to analyse the specific risk factors, clinical characteristics, and outcomes of *Cutibacterium* spp. IASIs compared with IASIs caused by other microorganisms and to analyse the impact of the use of rifampin, depending on the surgical strategy.

## 2. Results

### 2.1. Study Population

In total, 411 patients were included in this study. The *Cutibacterium* group included 27 (6.57%) patients diagnosed with *Cutibacterium* spp. IASIs (25 *C. acnes*, 1 *Cutibacterium avidum*, and 1 *Cutibacterium granulosum*), and the non-*Cutibacterium* group included 384 patients. Overall, 33% (135/411) were polymicrobial IASIs; there were 123 of 384 (32%) non-*Cutibacterium* IASIs vs. 12 of 27 (40.7%) *Cutibacterium* spp. cases, with coagulase-negative staphylococci (*n* = 4) and *Staphylococcus aureus* (*n* = 3) being the most frequent microorganisms involved. It should be noted that the percentage of polymicrobial infections in the *Cutibacterium* group was higher among early infections, 10/20 (50%), than among late infections, where only 2/7 (28%) were polymicrobial (*p* = 0.408). Table 1 shows the microbiological diagnosis of the NC group.

The baseline demographics and surgical characteristics of the population are described in Table 2 (Appendix A includes only patients with monomicrobial infections).

Patients in the *Cutibacterium* group were significantly younger, tended to have lower BMIs, and had fewer comorbidities (Charlson comorbidity index of 0 in 77% of cases and an ASA index equal or less than 2 in 88.9%). The main reason that led to fusion surgery in both groups was degenerative disease, with more than 50% of the cases. In the *Cutibacterium* group, there were significantly more cases in which the operation occurred due to spinal deformity (29.6% vs. 13.5%, *p* = 0.022). Although surgery involved more fused segments in patients in the *Cutibacterium* group, there were no differences in terms of the surgical time. We also did not detect differences in terms of the anatomical region involved when we analysed the entire cohort together, but if we analysed only monomicrobial infections, there was a trend towards more thoracic fusions (Appendix A).

### 2.2. Infection Characteristics

The infection characteristics of patients with IASIs caused by *Cutibacterium* spp. vs. other pathogens are compared in Table 3 (Appendix A for monomicrobial infections).

IASIs were diagnosed significantly later in the *Cutibacterium* group than in the NC group (median time of 23 days (IQR: 9–176) vs. 13 days (IQR: 7–22), *p* = 0.025). We highlight that *Cutibacterium* spp. infections are clearly predominant in late IASIs, where it represents 16% of the cases (7/43) vs. only 5.4% of early IASIs (20/368) (*p* = 0.015).

Symptoms suggesting an acute surgical site infection (fever, local erythema or oedema, wound secretion, and dehiscence of surgical wound) were less frequent in patients with *Cutibacterium* spp. infections except for pain and sinus tract, which were more frequent. Regarding the analytical parameters, the only significant difference was lower C-reactive protein values in the *Cutibacterium* group.

### 2.3. Multivariate Regression Analysis of Risk Factors Associated with Cutibacterium *spp*. IASIs

A multivariate logistic regression analysis was performed to investigate the risk factors for IASIs caused by *Cutibacterium* spp. vs. other microorganisms. The variables included after univariate analysis were: age, BMI, and number of instrumented segments fused. A younger age was the only significant variable detected with an odds ratio of 1.04 (95% confidence interval 1.019–1.06), *p* < 0.001 (Table 4).

### 2.4. Treatment and Outcome

Overall, the surgical strategy used was debridement with implant retention in 354 (86.1%) cases (19/27 (70.4%) in the *Cutibacterium* group vs. 335/384 (87.2%) in the NC group) and implant removal without reimplantation in 57 (13.9%) cases (8 (29.6%) in the *Cutibacterium* group and 49 (12.8%) in the NC group), *p* = 0.014. *Cutibacterium* spp. infection cases treated with implant removal were seven late infections and one early polymicrobial cervical infection that required subsequent stabilisation surgery using a double approach.

Regarding antibiotic treatment, patients with *Cutibacterium* spp. infections received shorter antibiotic regimens (mean treatment duration of 60.3 days (SD: 22.6) vs. 72.5 days (SD: 53.9), *p*= 0.014). However, the duration of antibiotic treatments was not significantly longer in patients treated with implant retention (72.9 days (SD: 53.01) vs. 70.7 days (49.1)) compared to patients for whom implants were removed. In all cases, antibiotic treatment was adjusted according to the susceptibility pattern of the bacteria isolated in intraoperative cultures. *Cutibacterium* spp. infections were treated with clindamycin in 13 cases, penicillin in 11 cases, and linezolid in 3 cases. Rifampin was added to the basal treatment in 8 (30%) patients with a *Cutibacterium* spp. IASI for a median time of 52.5 days (IQR: 39–72.7). All patients treated with rifampin except one had an early infection and were treated with DAIR. Only one patient, with a chronic fistulised infection at the level of the lumbar spine, was treated with implant removal and a targeted antibiotic regimen that included rifampin. None of the patients with *Cutibacterium* spp. IASIs had treatment failure regardless of rifampin use.

## 3. Discussion

The present study confirms the previously reported trend of *Cutibacterium* spp. as a cause of an IASI in the younger population undergoing spinal surgery. Previously described factors, including a lower BMI and the instrumentation of the thoracic region, have not been confirmed in this study, but there were trends suggesting that they might be relevant.

Our study shows a low rate of IASIs due to *Cutibacterium* spp. (<10%), which is in line with previous publications [18,19]. The study with the highest number of *Cutibacterium* spp. infections (59) is a case–control study published by Grossi et al. [11], but they do not report the rate of *Cutibacterium* spp. IASIs in their entire cohort. Zhou et al. recently published a systematic review and meta-analysis on the incidence of IASIs [4], in which they estimate that around 50% of these infections are caused by *S. aureus*, but again they do not offer specific data on the incidence of *Cutibacterium* spp. infections. Some authors comment that *Cutibacterium* spp. infections might be underestimated when the pathogen is considered a contaminant or overestimated when a contaminant is considered a pathogen [1].

Despite the fact that most *Cutibacterium* spp. infections diagnosed in our series were early IASIs, these only accounted for 5% of the global early IASIs of our entire cohort, while the 7 cases of a late IASI represented 16% of the late ones, a fact that is in line with the literature which traditionally associates *Cutibacterium* spp. With late infections [20]. However, recent studies have shown that it can also be a cause of acute infections [11,18], which is in line with our results. Differentiation between early and late infections is important because symptoms may differ. Early infections are usually associated with wound healing and drainage problems and local symptoms, while late IASIs present with fistulisation or even without wound problems, only with pain and loosening of the implants. Early infections tend to be treated with implant retention, and late infections are more likely to undergo implant removal.

Our main finding is that younger age is clearly related to *Cutibacterium* spp. Infection, which is in line with previous publications [11]. The only work in which age was not related with *Cutibacterium* spp. Infection studied a population with a median age of 24 years, precluding an assessment of the effect of age [20]. The fact that in that study, 34 of 74 IASIs were caused by *Cutibacterium* spp. seems to support the notion that this microorganism is more frequent in younger populations. It has also been described that *Cutibacterium* spp. IASIs are associated with long fusions with the frequent involvement of the thoracic segments [11,21], probably because these are usually surgeries that include more levels, though we only observed such a trend in our subset of patients with monomicrobial infections. The upper back is one of the areas of the body that has been found to be disproportionately colonised by *Cutibacterium* spp. in microbiome studies due to the higher concentration of sebaceous glands, which are the preferred environment for this microorganism [8,22]. This would explain the higher incidence of infections when the thoracic region is involved, and we believe this could be a relevant factor even if we could not confirm it with our data. In addition, it is also likely that the younger population undergoes more frequent surgical interventions in the more colonised upper back region, mainly in cases of idiopathic scoliosis. The older population is more likely to undergo surgery in the lumbosacral region, where colonisation by *Cutibacterium* spp. is lower (Olsen, Wenzel). The current data are insufficient to support this hypothesis, but larger studies could investigate this matter further. The fact that patients at risk of infection by *Cutibacterium* spp. have some specific features suggests that tailored preventive measures for this population should likely be implemented. In adolescent patients operated on for idiopathic scoliosis, some authors have proposed different strategies such as the local administration of vancomycin powder before wound closure or treating patients with *C. acnes* lesions with tetracycline 2 weeks before surgery and during the first postoperative week [23]. Other authors have suggested switching antibiotic prophylaxis from cefazoline to cefamandole in the adolescent population [24]. However, these strategies have only been published in reference to experiences in single centres, and no high-quality studies have been conducted to confirm these recommendations.

The prognosis of an IASI caused by *Cutibacterium* spp. in our series was very good since none of the patients had treatment failure regardless of the surgical strategy (DAIR vs. implant removal) or whether rifampin had been associated with the antibiotic scheme. Similarly, the authors of a recently published study on *Cutibacterium* spp. periprosthetic joint infections were unable to demonstrate the benefit of rifampin use and therefore concluded that a rifampin combination is not markedly superior to single regimens without it [17]. The fact that, in our series, 66.7% did not have treatment failure without rifampin is in line with those findings. There does not seem to be enough evidence in favour of the use of rifampin in these scenarios of orthopaedic infections caused by *Cutibacterium* spp., but the numbers are still too low to draw definitive conclusions and more studies are needed.

In addition to the chosen surgical strategy and antibiotic scheme, the length of antibiotic treatment is also often an important factor for clinical success. Traditionally, better outcomes were suggested to occur for patients treated with debridement and implant retention, when long antibiotic courses followed by a suppressive antimicrobial therapy were used [13]. However, in our previously published research, shorter antimicrobial courses (4 to 8 weeks) were not inferior to more than 8 weeks of optimised antibiotics in patients with early IASIs [12,25]. These findings are confirmed in the present sub-analysis since none of the patients with *Cutibacterium* IASIs had treatment failure regardless of the duration of the antibiotic treatment administered.

This study has several strengths and limitations. It has been performed on the basis of our previous published cohort of IASIs, the largest to our knowledge, with the participation of 14 Spanish hospitals that have multidisciplinary teams specializing in osteoarticular infections, which is undoubtedly our main strength. On the contrary, the main limitation is the small number of *Cutibacterium* spp. cases included, precluding a deeper investigation regarding risk factors. Another limitation is that it is not possible to elucidate whether *Cutibacterium* spp. was a pathogen or just the result of contamination in the 11 cases of polymicrobial infection. Recent in vitro studies suggest that, under anaerobic conditions, polymicrobial infections could be due to a symbiotic relationship between C. acnes and staphylococci, and not the result of contamination [26]. The results of the comparison of monomicrobial infections, which are unlikely to be due to contamination, are presented in the Appendix A. Although the sample was smaller, the results are quite similar, with a younger age being the most relevant factor. Therefore, it is unlikely that contamination could have significantly influenced the results. Finally, the retrospective nature of the study precludes obtaining optimal quality data, as there is a higher risk of missing data than in prospective studies. Considerable effort was made to obtain the best available data and to minimise missing information at each centre, resulting in good-quality data overall.

## 4. Materials and Methods

### 4.1. Study Design, Setting, and Population

This is a cross-sectional study including data from a multicentre, retrospective, observational study performed in 14 Spanish hospitals enrolled in the GEIO group cohort [12]. The study was conducted according to the guidelines of the Declaration of Helsinki, and IRB approval was granted by the local Ethics Committee (PR298/18).

The aim of this study was to assess the specific characteristics of an IASI by Cutibacterium spp., in comparison with infections produced by other microorganisms. The secondary objectives were to establish the risk factors for infection by *Cutibacterium* spp. and the usefulness of adding rifampin to treatment in these patients.

Patients older than 16 years who presented with an IASI between January 2010 and December 2016, regardless of the time that had elapsed since instrumentation, were included. For cases to be included, the IASI had to be confirmed after surgical debridement and positive cultures. For the present study, we included all cases of an IASI diagnosed with *Cutibacterium* spp. with at least 2 positive cultures and compared them with all IASI cases caused by other microorganisms. IASIs not requiring surgical debridement, as well as cases in which infection was the primary indication for spinal instrumentation, were excluded. All cases included had a minimum follow-up of 12 months after the completion of antibiotic treatment.

### 4.2. Definitions

An IASI was diagnosed, as previously described [12], based on the presence of two mandatory criteria: (i) the presence of symptoms or signs compatible with a surgical site infection, such as localised pain or tenderness, purulent drainage from the incision or localised inflammatory signs, e.g., swelling, erythema, or local warmness; and (ii) intra-operative findings compatible with infection and/or the presence of positive intraoperative cultures for the same pathogenic microorganisms in at least two different diagnostic samples, including tissue biopsies for low virulent-species such as coagulase-negative staphylococci (CoNS) or *Cutibacterium* spp. In all cases, the surgical samples were subjected to prolonged incubation (up to 14 days).

As previously described [11,12], we distinguished between early infections and delayed–late infections based on whether they occurred within the first 3 months following the index procedure or were diagnosed afterwards.

To be considered healed, all patients had a minimum twelve-month follow-up period after the completion of the antibiotic treatment without treatment failure, as defined below.

Treatment failure was defined as either infection persistence, relapse, new infection, need for suppressive treatment because of the difficulty to control the infection, or death attributed to an IASI. Infection persistence or relapse was defined as proven when symptoms or signs of infection remained or reappeared once antibiotics had been stopped and there was a new identification of *Cutibacterium* spp.

### 4.3. Data Collection

Data collection included demographics, comorbidities, risk factors for SSIs, surgical data, infection treatment-related data (medical and surgical), including antibiotic treatments and duration, and treatment outcome (i.e., healing, relapse, or need for implant removal). Time of follow-up was at least one year after the conclusion of antibiotic treatment.

Patients were divided into two groups:

1. *Cutibacterium* group: IASIs caused by *Cutibacterium* spp.

2. Non-*Cutibacterium* (NC) group: IASIs caused by other microorganisms.

Both groups were compared with regard to their demographics, comorbidities, primary spinal disease, surgical history, and treatment. In addition, a subset analysis including only monomicrobial infections in both groups was performed and is presented as Appendix A.

### 4.4. Antibiotic Treatment

We analysed the evolution of our patients based on the duration of antibiotic treatment and the efficacy of adding rifampin to the main treatment for at least 2 weeks. The antibiotic treatment duration was calculated as the total duration for all drugs combined (intravenous (IV), oral (PO), and rifampin use).

### 4.5. Statistical Analysis

Categorical variables are described by counts and percentages, while means and SDs or medians and IQRs are used to summarise continuous variables. Comparisons between groups were performed with either a chi-square test or Fisher’s exact test for categorical variables, and the *t*-test or Mann–Whitney U-test for continuous variables, depending on if the variables followed a normal distribution. A Bonferroni correction was applied to avoid type I error while performing multiple comparisons. A *p*-value of 0.05 was considered statistically significant. After univariate analysis, the associated factors of *Cutibacterium* spp. IASIs were determined by multivariate logistic regression analysis, entering in the model clinically relevant variables with *p*-values < 0.1. The goodness of fit was assessed using the Hosmere–Lemeshow test. Statistical analyses were performed using IBM SPSS Statistics for Windows, Version 21.0. (Armonk, NY: IBM Corp.)

## 5. Conclusions

*Cutibacterium* spp. is associated with early and late IASIs. Younger age is significantly associated, and it is likely that the instrumentation of the thoracic region may play a role. In our experience, the use of rifampin to improve the outcome in the treatment of an *Cutibacterium* spp. IASI is not relevant since, in our series, none of the cases had therapeutic failure regardless of the use of rifampin. Future studies with a larger number of patients included are needed to support our conclusions and to better investigate the specific characteristics of *Cutibacterium* spp. IASI.

## Figures and Tables

**Table 1 antibiotics-12-00518-t001:** Microbiological findings of the non-Cutibacterium group.

	All Isolates
Gram-positives	277 (45.9%)
*Staphylococcus* spp.	191 (31.7%)
*S. aureus*	124 (20.6%)
Coagulase-negative staphylococci	67 (11.1%)
Gram-negatives	321(53.2%)
Enterobacteriaceae	243 (40.3%)
*P. aeruginosa*	62 (10.3%)
Polymicrobial infections	135 (32.9%)

Table 1 summarises microbiological findings from the non-Cutibacterium group. There were 577 isolates from 384 cases.

**Table 2 antibiotics-12-00518-t002:** Baseline demographics and surgical characteristics of the population.

**Baseline Demographics**			
	**Non-*Cutibacterium*** ** *n* ** **= 384**	***Cutibacterium* spp.** ** *n* ** **= 27**	** *p* ** **-Value**
Age	57.5 (17.9)	42.2 (22.9)	0.001
Median age and IQR	61.4 (47.0–71.5)	38.0 (19.1–61.1)	0.006
BMI	28.6 (6.2)	25.9 (5.1)	0.05
Gender (female)	212 (55.2%)	10 (37.0%)	0.067
Corticoid treatment	23 (6.0%)	0 (0%)	0.191
Previous surgery	23 (7.3%)	4 (4.2%)	0.278
Charlson comorbidity index = 0	191 (50.8%)	21 (77.8%)	0.007
ASA=>2	352 (94.9%)	18 (66.7%)	0.001
**Surgical Characteristics**			
	**Non-*Cutibacterium*** ** *n* ** **= 384**	***Cutibacterium* spp.** ** *n* ** **= 27**	** *p* ** **-Value**
Emergency vs. elective	37 (9.7%)	2 (5.1%)	0.697
Fusion > 6 segments	65 (17.6%)	9 (33.3%)	0.043
Number of fused segments	4.09 (3.5)	5.07 (3.7)	0.164
Surgical time	242.9 (124.6)	237.1 (124.0)	0.84
Cervical	31 (8.1%)	4 (11.4%)	0.225
Thoracic	135 (35.2%)	10 (37.0%)	0.843
Lumbar	321 (83.6%)	21 (77.8%)	0.434

Numerical variables are expressed with mean values and standard deviations and categorical variables with number of cases and percentages. IQR = interquartile range, ASA = American Society of Anaesthesiologists. The ASA (American Society of Anaesthesiology) index is intended to assess a patient’s preanesthetic medical comorbidities. Its values range from 1 to 5; patients with a score of 1 are considered healthy. The Charlson Comorbidity Index (CCI) predicts 10-year mortality based on the presence of comorbidities. It is widely used as a marker to measure health status. Patients with a value of 0 are younger than 50 and have no comorbidities.

**Table 3 antibiotics-12-00518-t003:** Infection characteristics.

	Non-*Cutibacterium* *n* = 384	*Cutibacterium* spp. *n* = 27	*p*-Value
Mean time from surgery to infection (days)	148.2 (662.3)	530.7 (1269.9)	0.008
Median time from surgery to infection (days)	13 (7–22)	23 (9–176)	0.025
Infection within the first year	361 (94%)	21 (77.8%)	0.001
Infection within the first 90 days	348 (90.6%)	20 (74.1%)	0.007
C reactive protein (mg/L)	116.1 (110.4)	69.8 (33.1)	0.025
Erythrocyte sedimentation rate (mm/h)	66.3 (33.1)	66.5 (34.7)	0.593
Leucocytes (mm^3^)	10,946.1 (4845)	10,349.6 (3474.7)	0.664
Wound dehiscence	179 (46.9%)	11 (40.7%)	0.538
Wound drainage	291 (76.4%)	14 (53.8%)	0.01
Sinus tract	25 (6.5%)	3 (11.1%)	0.361
Fever > 38 °C	169 (44.5%)	8 (30.8%)	0.173
Erythema, swelling	169 (44.4%)	8 (29.6%)	0.136
More than one debridement needed	24 (9.3%)	0 (0%)	0.217
Outcome (absence of treatment failure)	341 (89.0%)	27 (100%)	0.069
Implant removal	49 (12.8%)	8 (29.6%)	0.014

Numerical variables are expressed with mean values and standard deviations (or as median and interquartile range (IQR) when indicated). Categorical variables are expressed with number of cases and percentages.

**Table 4 antibiotics-12-00518-t004:** Multivariate regression analysis.

	Regression Coefficient B	Standard Error	*p*-Value	Odds Ratio	95% CI
Age	−0.042	0.015	<0.001	0.959	0.938–0.981
BMI	−0.016	0.043	0.710	0.984	0.904–1.071
Number of fused segments	−0.008	0.054	0.884	0.992	0.892–1.104

Hosmer–Lemeshow goodness-of-fit chi-square *p* = 0.515.

## Data Availability

The data presented in this study are available on request from the corresponding authors. The data are not publicly available due to institutional policy.

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
