# Peer review of "Cutibacterium spp. Infections after Instrumented Spine Surgery Have a Good Prognosis Regardless of Rifampin Use: A Cross-Sectional Study"

_antibiotics, 2023, doi:10.3390/antibiotics12030518_

Round 1
Reviewer 1 Report
Please provide the inclusion and exclusion criteria for patient selection.
Has the author taken ethical clearance? If yes then provide an ethical clearance number.
Conclusion must be improved. Authors say that "All cases in this series were cured without relapse, regardless of rifampicin use", What is the significant and intimation of the study?
-------------
Author Response
Please, see attachment

Reviewer 2 Report
This original article entitled "Cutibacterium spp. infections after instrumented spine surgery have a good prognosis regardless of rifampin use" written by Núñez-Pereira et al. described and compared the clinical outcomes of Infection after spinal instrumentation (IASI) with or without Cutibacterium spp.
In this 6-year observational, multicenter, retrospective study, the authors found that Cutibacterium spp. IASI (CG) had a better outcome regardless of rifampin use. The manuscript is well-written and such a concept is rarely introduced into the literature. With some improvement, this manuscript will fit the current literature.
# The title is well descriptive and the authors need to insert the nature of the study. For example, Cutibacterium spp. infections after instrumented spine surgery have a good prognosis regardless of rifampin use: a cross-sectional study.
#The abstract is nicely written and concisely explained all sections. However, in this part, the authors need to update the conclusion and future direction part.
#This manuscript is well introduced and the authors first described the surgical site infections, then moved to IASI and the involvement of Cutibacterium spp in these infections and its clinical relevance. Then the authors mentioned the main research question of how CG could clinically differ from NCG and the indication of using Rifampin in such cases. In this part, I missed a few sentences about the microbiology of Cutibacterium spp. I think this will hugely enrich the introduction.
# The authors may also need to mention the differences between early and late IASI in a few words. This is important for readers from non-clinical backgrounds. Though it is mentioned in the materials and methods section.
# The materials and methods section is well written and I have no issues with this part except that study design the authors should mention cross-sectional as a general category for the study design.
# Further, the authors did not explain the period for defining the treatment failure.
# In data collection, I guess authors need to specify that non-Cutibacterium (NC) group IASIs caused by other microorganisms as non-culturable other microorganisms.
# In the results section, I suggest that authors add the microbes of the NC group to a table as a supplementary file.
# The authors need to mention more about the Charlson score and ASA score either in the footnote of Table 1 or in the result.
# Table 2 needs to be revised from the English perspective.
# Could you add a table for multivariate analysis?
# No major issues in the discussion as it is well written and the authors smoothly moved from one section to another one.
# It would be interesting as there are only a few studies that reported Cutibacterium surgical infection to be summarized in a table with the year of publication, country, prevalence rate, and other information that authors think is essential to be included.
# As the author did not specify for each case whether it is an infection or contamination, and I know it is difficult to be done with such a methodology, thus, the author should comment on that in the limitation part.
# In the conclusion, the authors need to add a sentence about the future direction.
All the best
Author Response
Please, see attachment

Reviewer 3 Report
1- Abstract is long with many details of result, while in same time conclusion is very short, paraphrase them.
2- Chart with bars will be very useful to illustrate this article findings.
Author Response
Please, see attachment

Reviewer 4 Report
To the authors,
Núñez-Pereira et al. conducted a retrospective study titled “Cutibacterium spp. infections after instrumented spine surgery have a good prognosis regardless of rifampin use ”. The authors present interesting and important data on specific risk factors, clinical characteristics and outcomes of Cutibacterium IASIs compared with infections caused by other pathogens. Although well-structured and of great importance, the article has some shortcomings:
Title:
- I suggest rephrasing the title. The present study’s main objective was not to investigate the efficacy of rifampicin in IASIs, but rather to analyze specific risk factors, clinical characteristics and outcomes.
Methods:
- Please state the primary and secondary outcomes in the methods.
- 4.4: Please clarify if rifampicin was used as mono- or combination therapy.
Results:
- Table 1: Although not significant, there was an absolute difference of almost 20% between male and female patients in the Cutibacterium group. I suggest discussing this, since previous reports have shown sex-specific differences ( https://doi.org/10.1371/journal.pone.0202639)
- Table 2: Consider changing “Outcome (healed vs. not)” to “treatment failure”, to match definitions in the methods section.
Discussion:
- Line 194: Do the authors have an explanation as to why younger age may be related to Cutibacterium infections?
Minor comments:
- In the introduction, the authors may consider stating that Rifampicin is usually only prescribed in combination with other antibiotics rather than as monotherapy when treating Cutibacterium infections.
- I suggest referring to either “Rifampicin” or “Rifampin” uniformly throughout the manuscript.
- Table S2, “Outcomes”: Should this read “not healed vs. healed”? Similar to Table 2, I suggest rephrasing to “treatment failure”
- Discussion: Lines 243-247 are worded identically to the discussion found in “DOI: 10.1093/cid/ciaa1839“. Please rephrase.
Author Response
Please, see attachment

Round 2
Reviewer 2 Report
The authors responded well to all previous comments and I don't have further comments.